# Stichoposide C and Rhizochalin as Potential Aquaglyceroporin Modulators

**DOI:** 10.3390/md22080335

**Published:** 2024-07-25

**Authors:** Ji Woo Im, Ju Hyun Lim, Valentin A. Stonik, Jong-Young Kwak, Songwan Jin, Minkook Son, Hae-Rahn Bae

**Affiliations:** 1Department of Physiology, Dong-A University College of Medicine, Busan 49201, Republic of Korea; imk4130@gmail.com (J.W.I.); applejunta@hanmail.net (J.H.L.); physionet@dau.ac.kr (M.S.); 2G.B. Elyakov Pacific Institute of Bioorganic Chemistry, Far-Eastern Branch of the Russian Academy of Science, 690022 Vladivostok, Russia; stonik@piboc.dvo.ru; 3Department of Pharmacology, School of Medicine, Ajou University, Suwon 16499, Republic of Korea; jykwak@ajou.ac.kr; 4Department of Mechanical Engineering, Tech University of Korea, Siheung-si 15073, Gyeonggi-do, Republic of Korea; songwan@tukorea.ac.kr

**Keywords:** aquaglyceroporin, stichoposide C, peracetyl aglycon of rhizochalin, stopped-flow light scattering, glycerol permeability

## Abstract

Aquaporins (AQPs) are a family of integral membrane proteins that selectively transport water and glycerol across the cell membrane. Because AQPs are involved in a wide range of physiological functions and pathophysiological conditions, AQP-based therapeutics may have the broad potential for clinical utility, including for disorders of water and energy balance. However, AQP modulators have not yet been developed as suitable candidates for clinical applications. In this study, to identify potential modulators of AQPs, we screened 31 natural products by measuring the water and glycerol permeability of mouse erythrocyte membranes using a stopped-flow light scattering method. None of the tested natural compounds substantially affected the osmotic water permeability. However, several compounds considerably affected the glycerol permeability. Stichoposide C increased the glycerol permeability of mouse erythrocyte membranes, whereas rhizochalin decreased it at nanomolar concentrations. Immunohistochemistry revealed that AQP7 was the main aquaglyceroporin in mouse erythrocyte membranes. We further verified the effects of stichoposide C and rhizochalin on aquaglyceroporins using human AQP3-expressing keratinocyte cells. Stichoposide C, but not stichoposide D, increased AQP3-mediated transepithelial glycerol transport, whereas the peracetyl aglycon of rhizochalin was the most potent inhibitor of glycerol transport among the tested rhizochalin derivatives. Collectively, stichoposide C and the peracetyl aglycon of rhizochalin might function as modulators of AQP3 and AQP7, and suggests the possibility of these natural products as potential drug candidates for aquaglyceroporin modulators.

## 1. Introduction

Marine natural products are a rich source of potential drugs because of their high biodiversity, low toxicity, suitability for oral applications, and a variety of biological actions [1,2,3]. According to the global marine pharmaceutical pipeline website (www.marinepharmacology.org) (accessed on 20 April 2024), 15 marine-derived drugs have been approved by the United States Food and Drug Administration (US FDA), and 32 marine-derived compounds are currently in various phases of clinical trials for drug development. Most have been developed or are under development as anticancer drugs, whereas a few are targeted to diseases other than cancer, such as pain, hypertriglyceridemia, Alzheimer’s disease, and viral infections. Considerable scientific and technological advances in analytical methods and functional assays in the past decade have opened up new opportunities for the exploration of marine natural products as novel potential therapeutic agents for more diverse disease entities [4,5,6].

Membrane transport proteins are important targets for drug discovery and delivery [7,8]. Transport proteins are essential for translocating solutes across the plasma or intracellular membranes, thus maintaining homeostasis. Transport proteins include channels and membrane transporters that are divided into three distinct classes: solute carriers (SLC), ATP-binding cassette (ABC) transporters, and ATPase ion pumps [9,10]. According to the data from The HUGO Gene Nomenclature Committee (https://genenames.org/) (accessed on 1 May 2024), transport proteins account for approximately 6% of all human protein-coding genes: 427 genes of the SLC superfamily, 312 genes of the ABC transporters, 125 genes of ATPase, and 291 genes of channels. The loss or alteration of the function of membrane transporters and channels causes a wide variety of human diseases, including channelopathies [11,12,13]. Membrane transporters are rapidly emerging as potential drug targets but have remained understudied until recently [14,15]. Moreover, since membrane transporters function as drug transporters, they play key roles in the absorption, distribution, and elimination of drugs and determine the therapeutic efficacy and adverse reactions of drugs in the process of drug discovery and development [16,17,18].

Aquaporins (AQPs) are a family of small integral membrane proteins that primarily transport water across cell membranes along osmotic gradients. To date, 13 AQP subtypes have been found in mammals (AQP0-12), some of which permit the transcellular passage of glycerol and urea as well as water (AQP3, 7, 9, and 10), and are thus called aquaglyceroporins [19]. AQPs are key players in maintaining a normal water balance and energy homeostasis in the body [20,21]. AQPs have been implicated in multiple disorders including diabetes insipidus, brain edema, cancer, obesity, cataracts, and neuromyelitis optica [22,23,24,25,26]. In addition to AQP-based diseases, AQP modulators have potential therapeutic utility in various pathological conditions for correcting abnormal transepithelial fluid and glycerol transport [20,21,27]. Notwithstanding the vigorous efforts to identify aquaporin modulators, limited progress has been made in the development of aquaporin-based therapeutics [28]. The relatively low hit rate, poor druggability due to toxicity, and lack of specificity makes the exploration of new AQP-targeted drugs challenging [29,30]. Because marine natural products not only possess greater structural and chemical diversity but also less toxicity than synthetic chemical libraries, the difficulty in finding promising lead compounds as AQP modulators using synthetic compound libraries encouraged us to conduct the present study to screen the marine natural products for novel drugs targeting AQPs.

## 2. Results

### 2.1. Effects of Marine Natural Compounds on the Osmotic Water Permeability of the Erythrocyte Membrane

We first investigated whether marine natural compounds could alter the osmotic water permeability of mouse erythrocyte membranes. To measure the membrane water permeability, the kinetics of cell volume change were examined using light scattering (Figure 1a). In the presence of an inwardly directed gradient of an impermeable solute, water moves out of the cells according to the osmotic gradient through AQPs. Consequently, the cell volume decreases, which causes an increase in light scattering. The AQP inhibitors prevented water efflux and cell shrinkage, retarding the increase in light scattering. As shown in Figure 1b, in response to an inward gradient of 300 mM sucrose, the scattered light intensity rapidly increased and reached a steady state. HgCl_2_ is a well-known inhibitor of various AQPs and prevents cell shrinkage in response to osmotic gradients by inhibiting water efflux [31,32]. DMSO also slows osmotic water movement owing to an osmotic clamping effect rather than a direct inhibition of water efflux [33]. The value of the osmotic water permeability coefficient (P_f_), which was calculated from the time course of the increase in light scattering upon exposure to 300 mM sucrose, was 0.033 ± 0.004 cm/s (Figure 1c). In the presence of DMSO or HgCl_2_, the P_f_ value decreased by 30% and 88%, respectively. Figure 1d,e show the representative light scattering traces for 24 marine natural compounds (100 nM) as well as those of stichoposide C, rhizochalin, and their derivatives in response to 300 mM sucrose. Though the late plateau phase of the traces after 0.2 sec was changed by several compounds, the initial slope of the rising phase was not affected and thus the calculated P_f_ values showed no statistical difference when compared with that of sucrose alone, suggesting no significant effects of these natural compounds on the osmotic water permeability.

### 2.2. Effects of Marine Natural Compounds on Glycerol Permeability of Erythrocyte Membrane

We next investigated whether any marine natural compounds might change the glycerol permeability of the mouse erythrocyte membrane. To measure the membrane glycerol permeability, changes in the cell volume in response to an imposed glycerol gradient were measured by light scattering (Figure 2a). In the presence of an inwardly directed gradient of glycerol, a permeable solute, a water efflux through AQPs occurs first according to the osmotic gradient, followed by a water influx accompanying a glycerol influx through aquaglyceroporins. The aquaglyceroporin inhibitors prevented the late re-swelling phase, retarding the decrease in light scattering. As shown in Figure 2b, scattered light intensity rapidly increased for the first ~0.3 s in response to an inwardly directed gradient of 300 mM glycerol in the same manner as that of sucrose, but slowly decreased afterwards as the cell volume was recovered by the entry of glycerol and water. DMSO slowed the initial increase in light scattering but did not affect the late decreasing phase. However, HgCl_2_, an inhibitor of AQPs including aquaglyceroporins, reduced not only the initial rise but also the late decreasing phase of light scattering. Figure 2c shows the representative light scattering traces for 24 marine natural compounds (100 nM) in response to 300 mM glycerol. Though several marine natural compounds increased or decreased the decrement rate of the late phase of light-scattering kinetics driven by a glycerol influx through aquaglyceroporins, the calculated glycerol permeability coefficient (P_glycerol_) values showed no statistical difference when compared with that of sucrose alone.

### 2.3. Effects of Stichoposide C, Rhizochalin, and Their Derivatives on Glycerol Permeability of Erythrocyte Membrane

Among the marine natural products tested, stichoposide C, rhizochalin, and their derivatives most notably affected the decrement rate of the late phase of light-scattering kinetics driven by a glycerol influx through aquaglyceroporins. As shown in Figure 3, stichoposide C increased the glycerol permeability, whereas rhizochalin decreased it. The effect of stichoposide C on the glycerol permeability of erythrocyte membranes was further investigated. The promoting effect of stichoposide C on the glycerol permeability was concentration-dependent in the range from 100 nM to 1 μM (Figure 3b). In contrast to stichoposide C, stichoposide D did not exert any significant influence on the late phase of light-scattering kinetics up to a concentration of 1 μM (Figure 3a).

The effects of rhizochalin and its derivatives on the glycerol permeability of erythrocyte membranes were also investigated (Figure 3c). The peracetyl aglycon of rhizochalin decreased the glycerol permeability more profoundly than rhizochalin. However, the aglycon of rhizochalin, aglycon of Rhizochalin A, and rhizochalin peracetate did not have any significant effect on the glycerol permeability, in contrast to rhizochalin. Rhizochalin exhibited a concentration-dependent inhibition of glycerol permeability (Figure 3d).

The glycerol permeability coefficients (P_glycerol_) were calculated from the time course of the late decreasing phase of light scattering (Figure 4). The P_glycerol_ values in the presence of 300 mM glycerol alone or with HgCl_2_ were (3.71 ± 0.511) × 10^−6^ cm/s and (0.78 ± 0.134) ×10^−6^ cm/s, respectively. The P_glycerol_ values in the presence of stichoposide C or Stichoposide D were (10.71 ± 0.803) × 10^−6^ cm/s (2.88-fold increase, *p* < 0.01) and (4.64 ± 0.388) × 10^−6^ cm/s (1.25-fold increase), respectively. The P_glycerol_ values in the presence of rhizochalin or the peracetyl aglycon of rhizochalin were (1.33 ± 0.617) × 10^−6^ cm/s (2.79-fold decrease, *p* < 0.05) and (0.15 ± 0.013) × 10^−6^ cm/s (24.73-fold decrease, *p* < 0.01), respectively, indicating that the peracetyl aglycon of rhizochalin was a more potent inhibitor of glycerol permeability than that of rhizochalin.

### 2.4. Expression of AQP Subtypes in Mouse Erythrocytes

To elucidate which AQP subtypes are responsible for the water and glycerol permeability, we analyzed CD1 mouse erythrocytes in the blood vessels by immunohistochemistry using nine AQP subtypes. As shown in Figure 5a, AQP1 and AQP7 were strongly expressed in mouse erythrocyte membranes. We verified this finding with an immunofluorescence analysis using glutaraldehyde-fixed erythrocyte suspensions. Though a weak AQP9 signal was detected, AQP7 exhibited a strong expression in erythrocyte membranes comparable to AQP1 (Figure 5b). These results indicated that AQP7 was the main aquaglyceroporin that determined the glycerol permeability in CD1 mouse erythrocytes.

### 2.5. Effects of Stichoposide C, Rhizochalin, and Their Derivatives on AQP3-Mediated Transepithelial Glycerol Transport

The effects of stichoposide C and rhizochalin on aquaglyceroporins were further explored using AQP3-expressing HaCaT cell monolayers to directly measure transepithelial glycerol transport (Figure 6a). When cells grown on permeable supports were exposed to 20 mM glycerol in the lower basolateral chamber, the glycerol concentrations in the upper apical chambers increased with time up to 30 min in both YFP-hAQP1 and YFP-hAQP3 HaCaT cells (Figure 6b). However, glycerol concentrations in YFP-hAQP3 HaCaT cells were over two-fold higher than those in YFP-hAQP1 HaCaT cells at every measured time point, which validated our method to measure AQP3-mediated transepithelial glycerol transport using YFP-hAQP3 HaCaT cells. Stichoposide C significantly stimulated AQP3-mediated basolateral glycerol transport in YFP-hAQP3 HaCaT cell monolayers (Figure 6c,d). The glycerol concentrations in the upper apical chambers at 10 min and inside the cells at 30 min were 2.30-fold (*p* < 0.01) and 1.97-fold (*p* < 0.05) increased, respectively, by Stichoposide C. The peracetyl aglycon of rhizochalin significantly decreased the glycerol concentrations in the upper apical chambers at 10 min (1.63-fold, *p* < 0.05) and inside the cells at 30 min (1.52-fold, *p* < 0.05). Stichoposide D increased, and rhizochalin decreased AQP3-mediated transepithelial glycerol transport to some extent, but there were no statistically significant differences.

## 3. Discussion

In this study, we aimed to identify potential AQP modulators in marine natural products. Although none of the marine natural compounds tested had significant effects on the osmotic water permeability, we found several compounds that significantly affected the glycerol permeability in mouse erythrocytes. We confirmed the modulatory effects of stichoposide C and rhizochalin on aquaglyceroporins through transepithelial glycerol transport assays using AQP3-expressing human keratinocytes.

Natural products, including plants, animals, and minerals, continue to be the best sources of novel drugs owing to their structural diversity and biodiversity [5,6]. In 2023, ten new natural products or their direct derivatives were drug approved by the US FDA, which corresponds to 18% of a total of 55 new drugs on the market [34]. Because the total number of species and biochemical diversity in oceans is higher than that on land, marine natural products are more invaluable sources for potential drug discovery than terrestrial products [3,4]. Moreover, natural products released into the water are rapidly diluted and, therefore, need to be highly potent to have any effect. The advantages of marine natural products, along with the growing appreciation for functional assays and phenotypic screening, may further contribute to the revival of interest in natural products for drug discovery and development [35,36].

Triterpene glycosides are widely distributed not only in plants, but also in marine invertebrates, especially echinoderms, octocorals, and sponges [37,38]. Many investigators have attempted to develop marine triterpene glycosides as candidate anticancer agents based on in vitro and in vivo studies [39,40,41]. Stichoposides C and D are triterpene glycosides extracted from sea cucumbers of the family *Stichopodidae* [42,43]. Stichoposide C is a quinovose-containing hexoside, whereas stichoposide D is a glucose-containing hexoside (Figure 7). Stichoposide C is an active membrane-acting agent with anticancer activity, and is more potent than stichoposide D [44,45]. In the present study, stichoposide C strongly stimulated the glycerol permeability in murine erythrocytes and AQP3-expressing human keratinocytes, whereas stichoposide D did not. Based on the notion that a linear tetrasaccharide fragment in triterpene glycosides is essential for the actions leading to the modification of the cellular membrane [39,46,47], having quinovose, rather than glucose, as a second monosaccharide unit in glycosides seems to be critical for interaction with 5(6)-unsaturated sterols of cellular membranes in contact with aquaglyceroporins or for direct interaction with channel proteins. Otherwise, STC-specific activation of the sphingomyelinase-ceramide pathway might account for the different effects on aquaglyceroporins [44]. However, further studies on the structure–activity relationship of these molecules are needed to improve the efficacy and safety of these compounds in activating aquaglyceroporins.

Rhizochalin is a marine two-headed sphingolipid-like natural product isolated from the sponge *Rhizochalina incrustata* [48,49]. These compounds differ from classical sphingolipids in the presence of polar groups at α,ω-positions, which contain a terminal methyl group instead of a hydroxymethyl group unlike sphingoid bases, thus representing a unique class of bipolar lipids [50,51]. Rhizochalin and its analogs have been reported to have antibacterial, antifungal, and cytotoxic activities [52,53,54,55]. In the present study, the peracetyl aglycon of rhizochalin exhibited the most potent inhibitory effect on the glycerol permeability in murine erythrocytes and human AQP3-expressing epithelial cells. It might be assumed that the absence of a glucopyranosyl sugar moiety on one polar end, together with the presence of three acetyl moieties on the other polar end of two-headed bipolar lipids (Figure 7), favors an intimate interaction with aquaglyceroporin or neighboring membrane lipids and regulatory proteins. The molecular mechanism underlying the inhibitory effect of the peracetyl aglycon of rhizochalin on aquaglyceroporins and the structure–activity relationship of rhizochalin analogs require further investigation.

Aquaglyceroporins mediate various physiological and pathophysiological processes [56,57,58]. Water-selective AQPs function primarily in water and salt homeostasis, whereas water/glycerol-transporting aquaglyceroporins are primarily involved in energy metabolism and lubrication. Because aquaglyceroporin dysfunction is implicated in various human diseases and symptoms, including obesity, nonalcoholic fatty liver disease, psoriasis, cancer, polyuria, and glyceroluria, functional modulators of aquaglyceroporins are promising therapeutic targets for a wide range of clinical conditions [24,59,60,61]. A few dozen aquaglyceroporin modulators have been reported to date, but none have been developed or are under development. Metal compounds, such as HgCl_2_ and *p*-chloromercuribenzene sulfonate (pCMBS), were first identified as AQP inhibitors, including aquaglyceroporins, and are still the most commonly used inhibitors in functional assays [32]. The gold compound Auphen ([Au(phen)Cl_2_]Cl) and its copper analog Cuphen ([Cu(phen)Cl_2_]) inhibited AQP3 and AQP7 [62,63]. Recently, several aquaglyceroporin inhibitors were identified by screening a commercially available library of small molecules: DFP00173 for AQP3, Z433927330 for AQP7, and HTS13286 for AQP9 inhibitors [64,65]. Natural compounds were also reported to have modulating effects on aquaglyceroporins which are usually from plants, such as glycolic acid, 18β-glycyrrhetinic acid, curcumin, chrysin, and phloretin [61]. However, the application of aquaglyceroporin modulators in clinical trials is still far from being implemented, possibly because of the lack of selective and potent modulators that can be administered in vivo.

In this study, we searched for potential AQP modulators among marine natural products and found that stichoposide C and the peracetyl aglycon of rhizochalin were potential aquaglyceroporin modulators. Since mouse erythrocytes express AQP7 and transepithelial glycerol transport was measured using human AQP3-expressing cells, it can be stated that stichoposide C and the peracetyl aglycon of rhizochalin function as modulators of AQP3 and AQP7. AQP3 activators may exert therapeutic effects on defects in the hydration, lubrication, and proliferation of the skin, bladder, vagina, and respiratory system, whereas AQP7 activators may be applicable as useful adjuvants for obesity treatment. In addition, AQP3 and AQP7 inhibitors are worthy of development as anticancer drugs. Further studies are needed to verify the therapeutic effects of stichoposide C and the peracetyl aglycon of rhizochalin in animal models of human diseases or in 3D bioengineered disease models.

## 4. Materials and Methods

### 4.1. Marine Natural Products

The marine natural products were provided by Dr. V. A. Stonik at the Pacific Institute of Bioorganic Chemistry, Far Eastern Branch of the Russian Academy of Science, Vladivostok, Russian Federation. The thirty-one compounds tested were extracted from marine natural products. The compounds were dissolved in phosphate-buffered saline (PBS) or dimethyl sulfoxide (DMSO) as 1–2 mM stock solutions, and diluted to working concentrations ranging from 100 nM to 1 μM.

### 4.2. Erythrocyte Preparation

Thirty CD1 mice, 6–8 weeks old, were used in this experiment. The mice were anesthetized with avertin (2,2,2-tribromoethanol-tert-amyl alcohol, 250 mg/kg intraperitoneally). Blood was collected from the inferior vena cava in heparinized tubes. Freshly obtained whole blood was centrifuged at 120× *g* for 10 min to remove the plasma and buffy coat. The erythrocytes were washed three times and suspended in PBS to adjust the final hematocrit to 2%. All animal protocols were approved by the Dong-A University Medical School Institutional Animal Care and Use Committee (DIACUC-10-6-1).

### 4.3. Stopped-Flow Light Scattering Measurements

The water and glycerol permeabilities were measured via light scattering using a stopped-flow apparatus (SFM-3, Biologic, Seyssinet-Pariset, France). The erythrocyte suspension was subjected to a 300 mM inwardly directed gradient of either sucrose or glycerol. The kinetics of decreasing cell volume were measured from the time course of 90° scattered light intensity at 530 nm [66]. The osmotic water and glycerol permeability coefficients were calculated from the time course of light scattering using the following equation:dV(t)/dt = P_f_ × SAV × MVW × (C_in_ − C_out_)
where V(t) is the relative red cell volume as a function of time, SAV is the red cell surface area-to-initial volume ratio, MVW is the molar volume of water (18 cm^3^/mole), and C_in_ and C_out_ (mole/cm^3^) are the initial concentrations of the intracellular and extracellular solutes, respectively. The average erythrocyte surface area and volume were 7.2 × 10^−7^ cm^2^ and 3.1 × 10^−11^ cm^3^, respectively [67].

### 4.4. Tissues Preparation and Immunohistochemistry

CD1 mouse kidneys were fixed in 10% neutral buffered formalin for 48 h, paraffin-embedded, and sectioned at 5 μm using a microtome (RM 2125RT, Leica, Wetzlar, Germany). Paraffin sections were deparaffinized with xylene, serially rehydrated in 100%, 95%, and 70% ethanol, and boiled for 15 min in citrate buffer (10 mM sodium citrate, 0.05% Tween 20, and pH 6.0) for antigen retrieval. After blocking with 5% bovine serum albumin for 2 h, the sections were incubated with the following primary antibodies overnight at 4 °C at a dilution of 1:300: anti-AQP1, AQP2, AQP3, and AQP4 (Merck Millipore, Billerica, MA, USA); anti-AQP5, AQP6, and AQP7 (Alomone Labs, Jerusalem, Israel); anti-AQP8 (Mybiosource, San Diego, CA, USA); and anti-AQP9 (Bioss, Woburn, MA, USA). Subsequently, the sections were incubated with secondary anti-rabbit horseradish peroxidase (HRP)-linked IgG (Dako, Glostrup, Denmark) for 40 min at 25 °C, and HRP activity was determined by adding DAB+ chromogen (Dako). The sections were mounted using Marinol Mount (Muto Pure Chemicals, Tokyo, Japan) solution and xylene (Muto Pure Chemicals) (1:1), and then visualized using a digital slide scanner (Pannoramic Midi, 3DHistech, Budapest, Hungary) at the Neuroscience Translational Research Solution Center (Busan, Republic of Korea).

### 4.5. Cell Cultures

HaCaT cells, an immortalized human keratinocyte line, were stably transfected with a plasmid encoding the yellow fluorescent protein (YFP) and either human AQP1 (YFP-hAQP1) or human AQP3 (YFP-hAQP3) (gifts from Dr. Alan Verkman at the University of California, San Francisco, CA, USA) and selected using G418 (Invitrogen, Thermo Fisher Scientific, Waltham, MA, USA). Cells were grown in a DMEM medium (Gibco, Thermo Fisher Scientific) supplemented with 10% fetal bovine serum (Hyclone^TM^, Cytiva, Logan, UT, USA), penicillin G (100 U/mL), and streptomycin (100 μg/mL) and maintained at 37 °C in a humidified 5% CO_2_ incubator.

### 4.6. Measurement of Transepithelial Glycerol Transport

YFP-hAQP1-HaCaT cells and YFP-hAQP3-HaCaT cells were seeded at a density of 2 × 10^4^ cells/well onto 6.5 mm, 0.4 μm pore size Transwell permeable supports (Corning Life Sciences, Tewksbury, MA, USA), cultured for 3–5 days to a confluent state, and serum-deprived for 2–3 days before the experiments. Once the cells reached a confluent state, the integrity of the cellular barriers was assessed by transepithelial electrical resistance (TEER) using an epithelial voltohmmeter. When the TEER values were above 200 Ω·cm^2^, transepithelial flux experiments were performed [68].

Glycerol transport across HaCaT cell monolayers was determined by measuring the glycerol concentrations of the culture medium in the upper and lower compartments, as well as in the Transwell inserts. The cells were pretreated with marine natural products or HgCl_2_ for 10 min, and 20 mM glycerol was added to the bath solution on the basal side of the cells. Aliquots (10 µL) of samples were collected from the medium of the opposite side from the glycerol addition at the indicated time points. The glycerol concentrations were measured using a glycerol assay kit (Sigma-Aldrich, Saint Louis, MO, USA) according to the manufacturer’s instructions.

### 4.7. Statistical Analysis

All the data were expressed as the mean ± standard deviation (SD). For data analysis of the P_f_ and P_glycerol_ values, Student’s *t*-test was used to compare the mean values between the two groups. For data analysis of transepithelial glycerol transport, one-way analysis of variance (ANOVA), Duncan’s multiple range test was employed to determine the statistical significance of differences between the groups. We used the Statistical Package for Social Science (SPSS) ver. 22.0 (SPSS Inc., Chicago, IL, USA) with a significance level set at *p* < 0.05.

## Figures and Tables

**Figure 1 marinedrugs-22-00335-f001:**
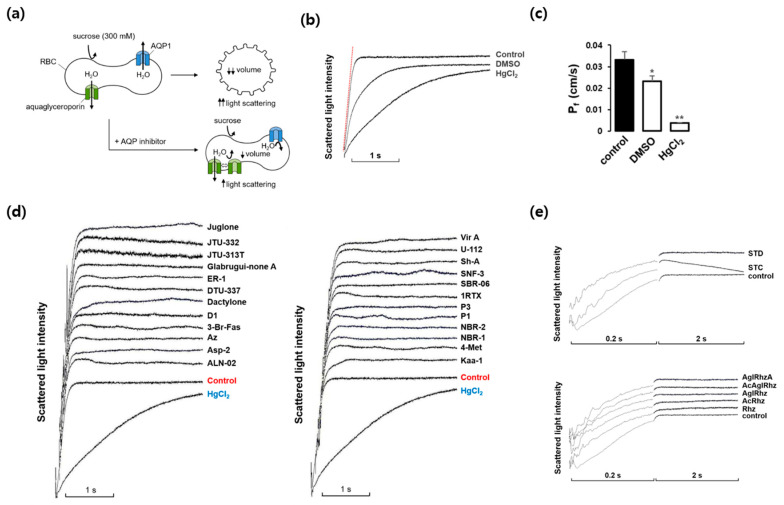
Effects of various marine natural products on osmotic water permeability in mouse erythrocytes. (**a**) Assay method. Erythrocytes were subjected to a 300 mM sucrose gradient. A rapid decrease of the cell volume due to the water efflux through AQPs increased light scattering intensity. AQP inhibitors prevented water efflux and cell shrinkage. (**b**) Osmotic water permeability (P_f_) was measured from the initial slope of the time course of scattered light intensity (a red line). Control (sucrose alone), DMSO (dimethylsulfoxide, 0.1%), and HgCl_2_ (0.5 mM). (**c**) The P_f_ values calculated from traces in (**b**). Values are the mean ± SD. * *p* < 0.05, ** *p* < 0.01. (**d**) The representative light scattering traces for 24 marine natural compounds (100 nM) in response to 300 mM sucrose. (**e**) The representative light scattering traces for stichoposide C (STC), stichoposide D (STD), rhizochalin (Rhz) and its derivatives in response to 300 mM sucrose. AglRhz, aglycon of rhizochalin; AcRhz, peracetyl rhizochalin; AcAglRhz, peracetyl aglycon of rhizochalin; AglRhzA, aglycon of rhizochalin A.

**Figure 2 marinedrugs-22-00335-f002:**
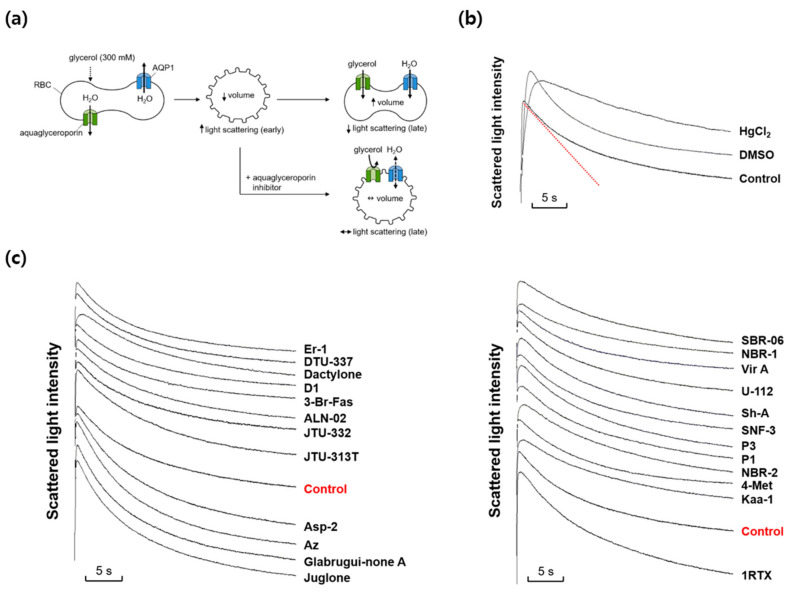
Effects of various marine natural products on glycerol permeability in mouse erythrocytes. (**a**) Assay method. Mouse erythrocytes were subjected to a 300 mM glycerol gradient. A rapid decrease of the cell volume (increased light scattering) due to the water efflux through AQPs was followed by cell re-swelling (decreased light scattering) due to glycerol and water influx. Aquaglyceroporin inhibitors prevented the late re-swelling phase. (**b**) Glycerol permeability was measured from the slope of the late decreasing phase of light scattering (a red line) in response to a 300 mM inwardly directed glycerol gradient (control). HgCl_2_ (0.5 mM); DMSO, dimethylsulfoxide (0.1%). (**c**) The representative light scattering traces for 24 marine natural compounds (100 nM) in response to 300 mM glycerol.

**Figure 3 marinedrugs-22-00335-f003:**
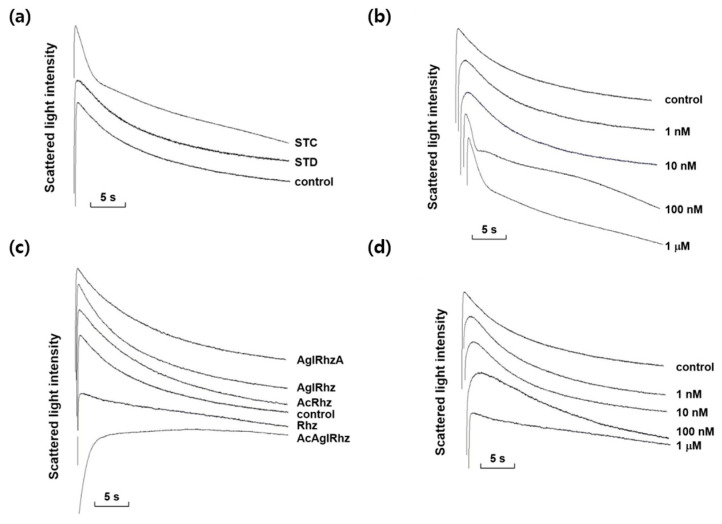
Effects of stichoposide, rhizochalin, and their derivatives on glycerol permeability in mouse erythrocytes. (**a**) The representative light scattering traces for stichoposide C (STC, 1 μM) and stichoposide D (STD, 1 μM) in response to 300 mM glycerol. (**b**) The representative light scattering traces for the absence (control) or presence of the indicated concentrations of STC in response to 300 mM glycerol. (**c**) The representative light scattering traces for 1 μM of rhizochalin (Rhz) and its derivatives in response to 300 mM glycerol. AglRhz, aglycon of rhizochalin; AcRhz, peracetyl rhizochalin; AcAglRhz, peracetyl aglycon of rhizochalin; AglRhzA, aglycon of rhizochalin A. (**d**) The representative light scattering traces for the absence (control) or presence of the indicated concentrations of Rhz in response to 300 mM glycerol.

**Figure 4 marinedrugs-22-00335-f004:**
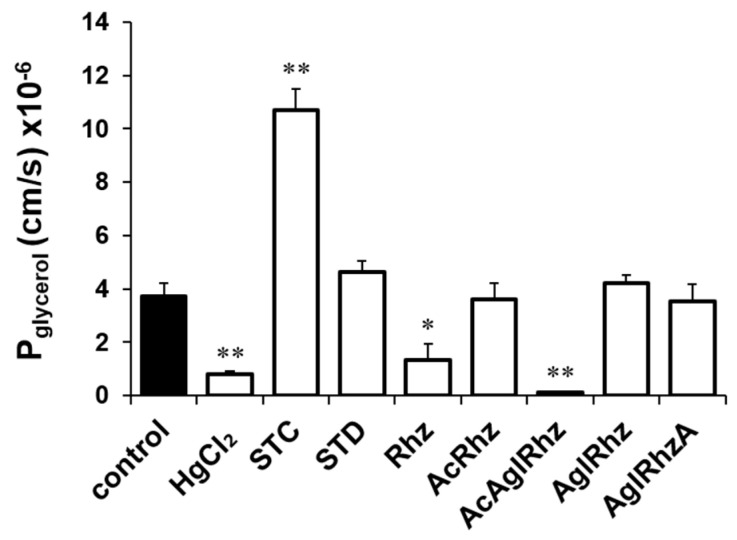
Glycerol permeability coefficients (P_glycerol_) of mouse erythrocytes. The P_glycerol_ value was calculated from the late decreasing phase of light scattering traces in the presence of 300 mM glycerol alone (control) or together with stichoposide (STC), rhizochalin (Rhz), and their derivatives (100 nM). STD, stichoposide D; AglRhz, aglycon of rhizochalin; AcRhz, peracetyl rhizochalin; AcAglRhz, peracetyl aglycon of rhizochalin; and AglRhzA, aglycon of rhizochalin A. Values are the mean ± SD (*n* = 20). * *p* < 0.05, ** *p* < 0.01 according to Student’s *t*-tests.

**Figure 5 marinedrugs-22-00335-f005:**
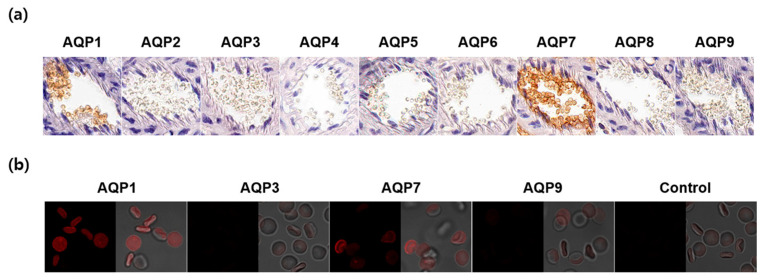
Expression of AQP subtypes in CD1 mouse erythrocytes. (**a**) Immunohistochemistry of paraffin-embedded kidney sections using anti-AQP1-AQP9 antibodies. (**b**) Immunofluorescence analysis of glutaraldehyde-fixed erythrocyte suspensions using anti-AQP antibodies. Control, secondary antibody alone.

**Figure 6 marinedrugs-22-00335-f006:**
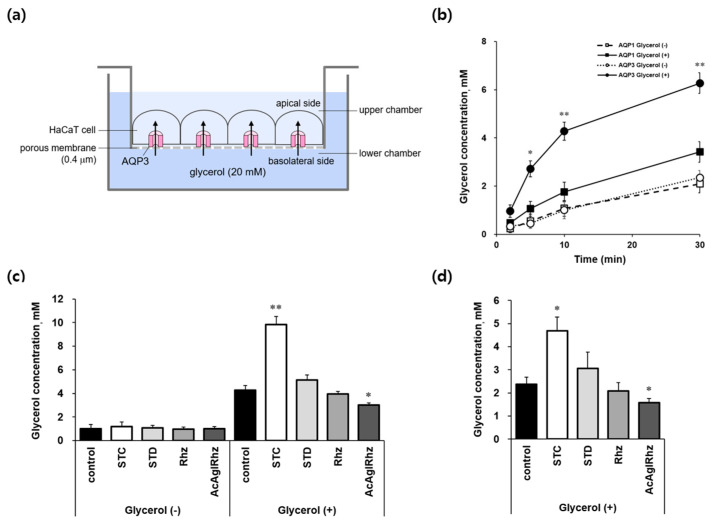
AQP3-mediated transepithelial glycerol transport in HaCaT cell monolayers. (**a**) Assay method. YFP-hAQP3 HaCaT cells grown on permeable supports were exposed to 20 mM glycerol in the lower chamber of the Transwell. The basal-to-apical glycerol transport across the cell monolayer was assessed by measuring the glycerol concentration in the upper chamber as well as inside the cells. (**b**) The glycerol concentrations of the upper chamber bathed on the apical side of YFP-hAQP1 and YFP-hAQP3 HaCaT cells at the indicated time points. (**c**) The glycerol concentrations of the upper chamber bathed on the apical side of YFP-hAQP3 HaCaT cells at 10 min. 1 μM of stichoposide C (STC), stichoposide D (STD), rhizochalin (Rhz), or peracetyl aglycon of rhizochalin (AcAglRhz). (**d**) The glycerol concentrations inside YFP-hAQP3 HaCaT cells at 30 min. Values are the mean ± SD (n = 10). * *p* < 0.05, ** *p* < 0.01 according to one-way ANOVA followed by Duncan’s test.

**Figure 7 marinedrugs-22-00335-f007:**
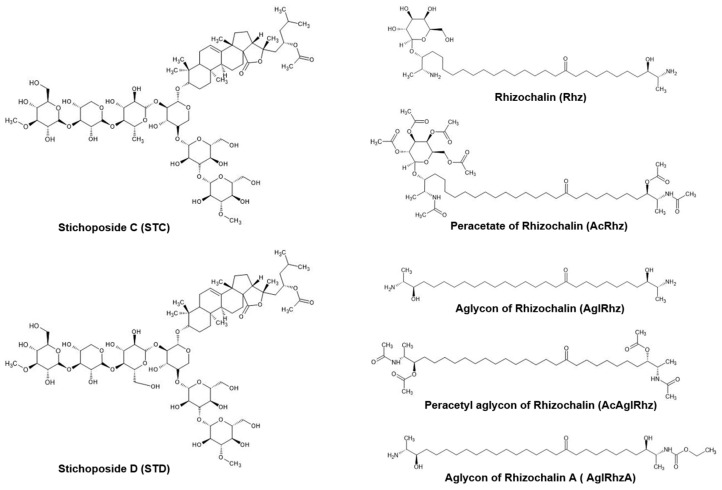
Structures of Stichoposide, Rhizochalin, and their derivatives.

## Data Availability

The data presented in this study are available on request from the corresponding author.

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
