# Peer review of "Stichoposide C and Rhizochalin as Potential Aquaglyceroporin Modulators"

_marinedrugs, 2024, doi:10.3390/md22080335_

Round 1
Reviewer 1 Report
Comments and Suggestions for Authors
The authors present a study on the effect of marine natural products on the water and glycerol permeability of mouse red blood cells. While the water permeability remained unaffected, the permeability for glycerol appeared increased or decreased depending on the applied compound.
Points to be addressed by the authors:
1. The term “transport” is used in connection with aquaporin permeability throughout the ms. AQPs are strict channels, though. Therefore, “transport” cannot be used.
2. Figure 3 shows biphasic water/glycerol permeability curves for series of compound concentrations. Since the study aims to deal with drug discovery, the dose-response should be plotted as sigmoidal curves using the Hill equation and IC50 values determined (incl. errors).
3. AQP7 is stated to be the prominent aquaglyceroporin in mouse erythrocytes based on antibody-staining. This notion is not compatible with earlier publications, which found AQP9 to be responsible for glycerol permeability in mouse red blood cells. Respective ko animals showed decreased erythrocyte glycerol permeability. The quality and selectivity of the antibodies is not shown, e.g. by Western blot. How reliable are the data?
4. Effects on AQP3 were tested using HaCaT cells in a monolayer, which is a quite different system compared to the erythrocytes used before. This may be highly relevant in view the putative mode of action of the compounds, see next points. For better comparison, human erythrocytes should be used, which contain AQP3 in the plasma membrane.
5. A discussion of the drug likeliness of the presented compounds should be added as their structures are far off from drug-like molecule structures.
6. From the shown structures and statements in the discussion, it seems that the used compounds exert membrane effects rather than direct protein interaction similar to the classical phloretin, which also acts indirectly on AQPs (among other membrane proteins). This should be elaborated.
Reviewer 2 Report
Comments and Suggestions for Authors
In this manuscript, following screening, the authors reported evidence of two marine natural products, i.e., stichoposide C and rhizochalingene, as modulators of aquaporins. While some interesting findings have been presented, careful recheck of the manuscript is needed - considering that some parts are somewhat confusing, as described below. Novelty of the findings should be clearly highlighted too, either in the last paragraph of INTRO or in DISCUSSION - including clarifying the novelty/rationale of studying the 24 selected natural products and not any other marine natural products. I have a few minor comments as listed below.
1. Please check that information mentioned should be supported by citing references where appropriate. For example, see statements in lines 90-93 and 309-312. Please recheck the draft again for this issue. The M&M also seems to have very few cited references. Can the authors recheck this?
2. The way the authors referred to figures seems somewhat confusing.
-For example, in line 96, Figures 1, 3a and 3d are already mentioned. But after Figure 1, in the draft, the authors showed Figure 2 instead of Figure 3. In this case, shouldn’t the positions of Figure 2 and Figure 3 be swapped?
-In line 151, the authors mentioned Figure 4. But after this line, Figure 4 is not presented. In fact, two lines after mentioning Figure 4, the authors discussed Figure 3e.
-Similarly, after mentioning Figure 5 in line 181, Figure 4 is shown.
-In section 2.2, the authors discussed results in Figure 2a, then they discussed results in Figure 4 (line 128). Not other subfigures of Figure 2 or Figure 3.
In short, it seems confusing, disorganized, and not easy to follow.
3. In lines 88-89, the authors wrote “The calculated Pf values in the presence of marine natural compounds were not changed …” – But based on the scattered light intensity results in Figure 1, it seems that at least some may show increases in Pf. Can the authors recheck? Or was the lack of change concluded due to lack of statistical significance?
4. In lines 107 (Figure 1 caption) and 140 (Figure 2 caption), “control” – does it refer to the absence of added natural compound?
5. In the initial screening, the authors used 100 nm as chosen concentration. What is the justification of this?
6. In Figure 1, since the scattered light intensity results have been obtained, why didn’t the authors show the Pf data derived?
7. Just like in Figure 1, the authors could consider showing the Pglycerol data for results derived from Figure 2c.
8. The caption of Figure 2 should be improved to indicate the full names of the abbreviations indicated in Figure 2c.
9. This sentence structure is confusing and can be revised appropriately – lines 146-147 - “In contrast to stichoposide C, stichoposide D, a structural analog of stichoposide C, did not …”.
10. To make Figure 4 easily understood as a standalone, the caption can be improved by indicating what statistical test was used, since the authors mentioned two types of statistical tests in their M&M.
11. Student’s t-test and one-way ANOVA are not a single test with two different names. So, it sounds unclear that in M&M, the authors wrote “Student’s t-test *or* one-way ANOVA” (lines 389-390). Please recheck. Perhaps the authors can briefly clarify in M&M which test was used for which set of data.
12. Importantly, the authors only mentioned using P < 0.05 as a threshold for statistical significance (line 390), so it might be good that the authors briefly clarify in the result description why they felt it important to highlight that in some cases/data P < 0.01 level was achieved.
13. The authors reported that AQP7 was the main aquaporin form. In this case, could the authors clarify why they tested with AQP3-expressing HaCat cells, but not AQP7-expressing HaCat cells?
14. Figure 6 caption – too long. I would suggest that some info, where relevant, should be moved to M&M instead.
15. In the section 4.7. Statistical analysis – the authors should indicate the statistical software they used.
